# Structural basis for potent neutralization of SARS-CoV-2 and role of antibody affinity maturation

Nicholas K. Hurlburt [1], Emilie Seydoux[1], Yu-Hsin Wan[1], Venkata Viswanadh Edara[2], Andrew B. Stuart[1], Junli Feng[1], Mehul S. Suthar[2], Andrew T. McGuire [1,3], Leonidas Stamatatos[1,3] & Marie Pancera [1,4 ✉]

SARS-CoV-2 is a betacoronavirus virus responsible for the COVID-19 pandemic. Here, we determine the X-ray crystal structure of a potent neutralizing monoclonal antibody, CV30, isolated from a patient infected with SARS-CoV-2, in complex with the receptor binding domain. The structure reveals that CV30 binds to an epitope that overlaps with the human ACE2 receptor binding motif providing a structural basis for its neutralization. CV30 also induces shedding of the S1 subunit, indicating an additional mechanism of neutralization. A germline reversion of CV30 results in a substantial reduction in both binding affinity and neutralization potential indicating the minimal somatic mutation is needed for potently neutralizing antibodies against SARS-CoV-2.

[1] Fred Hutchinson Cancer Research Center, Vaccines and Infectious Diseases Division, Seattle, WA, USA. [2] Division of Infectious Disease, Department of Pediatrics, Emory Vaccine Center, Yerkes National Primate Research Center, Emory University School of Medicine, Atlanta, GA, USA. [3] Department of Global Health, University of Washington, Seattle, WA, USA. [4] Vaccine Research Center, National Institutes of Allergy and Infectious Diseases, National Institute of Health, Bethesda, MD, USA. ✉email: mpancera@fredhutch.org

Coronavirus disease (COVID-19) was declared a pandemic in March 2020 by the World Health Organization[1]. As of 9 August 2020, there were ~19.5 million infections and over 728,000 deaths worldwide[2]. It is caused by a coronavirus of the β-family, named severe acute respiratory syndrome coronavirus 2 (SARS-CoV-2)[3], as it is closely related to SARS-CoV-1[4]. Their genomes share 80% identity and they utilize angiotensin-converting enzyme 2 (ACE2) as receptor for entry[5–11]. Viral entry depends on the SARS-CoV-2 spike glycoprotein, a class I fusion protein that comprised two subunits, S1 and S2. S1 mediates ACE2 binding through the receptor-binding domain (RBD), whereas the S2 subunit mediates fusion. Overall, the spike shares 76% amino acid sequence homology with SARS[4]. High-resolution structures of the SARS-CoV-2-stabilized spike in the prefusion revealed that the RBD can be seen in a up or down conformation[5,6]. It's been shown that some of the neutralizing antibodies bind the RBD in the up conformation similar to when the ACE2 receptor binds[12]. Currently, there is no vaccine available to prevent SARS-CoV-2 infection and highly effective therapeutics have not been developed yet either. The host immune response to this new coronavirus is also not well understood. We, and others, sought to characterize the humoral immune response from infected COVID-19 patients[12–14]. Recently, we isolated a neutralizing antibody, named CV30, which binds the RBD, neutralizes a pseudovirus with an IC50 of 0.03 μg/ml and competes binding with ACE2[15]. However, the molecular mechanism by which CV30 blocked ACE2 binding was unknown. Herein, we present the 2.75 Å crystal structure of SARS-CoV-2 RBD in complex with the Fab of CV30, assess the role of affinity maturation and determine in details its mechanism of neutralization.

## Results

**Structure of CV30 in complex with RBD.** The structure revealed that CV30 binds almost exclusively to the concave ACE2-binding epitope (also known as the receptor-binding motif (RBM)) of the RBD using all six complementary determining region (CDR) loops with a total buried surface area of ~1004 Å², ~750 Å² of which are from the heavy chain, and ~254 Å² are from the light chain (Fig. 1a, PDB: 6XE1, Supplementary Table 1). Twenty residues from the heavy chain and 10 residues from the light chain interact with the RBD, forming 13 and 2 hydrogen bonds, respectively (Fig. 1b, c and Supplementary Table 2). There are 29 residues from the RBD that interact with CV30, 19 residues with the heavy chain, 7 residues with the light chain, and 3 residues with both (Supplementary Table 2). Of the 29 interacting residues from the SARS-CoV-2 RBD, only 16 are conserved in the SARS-CoV-1 S protein RBD (Fig. 2c), which could explain the lack of cross-reactivity of CV30 to SARS-CoV-1 S as we observed previously by biolayer interferometry (BLI)[15] and to SARS-CoV-1 RBD by enzyme-linked immunosorbent assay (Supplementary Fig. 1).

**CV30 minimal affinity maturation plays a role in neutralization.** The CV30 heavy chain is minimally mutated with only a two-residue change from the germline and both of these residues (Val27-Ile28) are located in the CDRH1 and form nonpolar interactions with the RBD. We reverted these residues to germline to assess the role of affinity maturation to RBD-binding and neutralization potency. The germline CV30 (glCV30) antibody bound to RBD with ~100-fold lower affinity (407 nM) (Fig. 1d and Supplementary Table 3) compared to CV30 (3.6 nM[15]), which was largely due to a much faster off-rate for glCV30. glCV30 neutralized SARS-CoV-2 pseudovirus with ~350-fold less potency with an IC50 of 10.3 vs. 0.03 μg/mL for CV30 (Fig. 1e). We also performed live virus neutralization assay. Compared to

the pseudovirus assay, CV30 neutralized the live virus less potently, with an IC50 of 0.118 μg/mL, whereas glCV30 failed to achieve 50% neutralization (Fig. 1f).

Val27 forms a weak nonpolar interaction with the RBD Asn487 and sits in a pocket formed by CDRH1 and CDRH3. It is possible that Phe27, present in glCV30, could change the electrostatic environment, thus contributing to the lower affinity and neutralization. The Ile28 sidechain forms nonpolar interactions with the RBD Gly476-Ser447, particularly the $C_\gamma$ atom, which the glCV30 Thr28 would be incapable of making. Thus, minimal affinity maturation that occurred in CV30 significantly impacted the ability of this monoclonal antibody (mAb) to bind and neutralize SARS-CoV-2.

**CV30 neutralizes by blocking ACE2 binding and inducing S1 shedding.** We have shown previously that CV30 competes with ACE2 for binding to the RBD[15] and we therefore examined the structural mechanism of the receptor blocking by superimposing the SARS-CoV-2 RBD/ACE2 complex (PDB: 6LZG)[9] with the CV30 Fab/RBD complex. The structure of the RBD was used to align the two complexes and showed that CV30 binding did not induce any conformational changes in the RBD from the ACE2-bound complex. The aligned RBD had a RMSD of 0.353 Å over 166 $C_\alpha$ atoms. The structure reveals that the CV30 epitope overlaps almost completely with the ACE2 epitope. A total of 26 residues of the SARS-CoV-2 RBD interact with hACE2, CV30 binds to 19 of these residues (Fig. 2a), indicating that CV30 likely neutralizes the virus by preventing the binding of ACE2 to RBD by direct steric interactions. In addition, it was recently shown that ACE2 blocking antibodies could induce premature shedding of the S1 subunit as a mechanism of neutralization[16,17]. We incubated HEK-293-6E cells expressing full-length SARS-CoV-2 S isolate USA-WA1/2020 with saturating concentrations of CV30 or glCV30 and monitored the mean fluorescence intensity (MFI) of antibody binding over time by fluorescence-activated cells sorting (FACS)[16]. CV30 showed greatly decreased binding compared to CR3022 control, which has been shown to not induce shedding[18], indicating that CV30 binding leads to S1 dissociation (Fig. 2b). glCV30 showed a slight decrease in binding again indicating that minimal mutations have a large effect on the function of CV30.

**IGHV3-53*1 antibodies bind RBD in nearly identical manner.** Recently, the structure of two potent neutralizing anti-RBD antibodies were published, B38 and CB6[12,14]. CV30 shares a similar germline heavy chain V-genes but all three have diverse germline kappa V-genes (CV30 is IGKV3-20*01, B38 is IGKV1-9*01, and CB6 is IGKV1-39*01; Supplementary Fig. 2). Both CV30 and B38 use IGHV3-53*01, whereas CB6 uses IGHV3-66*01, which is only one amino acid different than 3-53*01 (Val12, which does not make contact with the epitope). CV30 and CB6 each have higher affinities, 3.6 and 2.5 nM, respectively, than B38, 70.1 nM[12,14,15]. Differences in affinity translate into differences in neutralization potency (the IC50s for CV30 and CB6 are 0.03 and 0.036 μg/mL, respectively, and that of B38 is 0.177 μg/mL). Interestingly, Thr28 was also mutated from germline to Ile in B38 but Phe27 was not. CB6 lacks both mutations found in CV30. Differences in other regions of the antibody, such as the CDRH3 and light chain are likely responsible for the overall potency all these antibodies (see below). To investigate the binding mechanism of the three antibodies, a superposition of the structures was created. All three bind in a nearly identical manner with the same angle of approach and similar footprints (Fig. 2c). The alignment of the Fv regions of B38 (PDB: 7BZ5)[14] and CB6 (PDB: 7C01)[12] to the Fv region of CV30 had a RMSD of 0.240 Å over 100 $C_\alpha$ atoms and 0.329 Å over 98 $C_\alpha$ atoms, respectively.

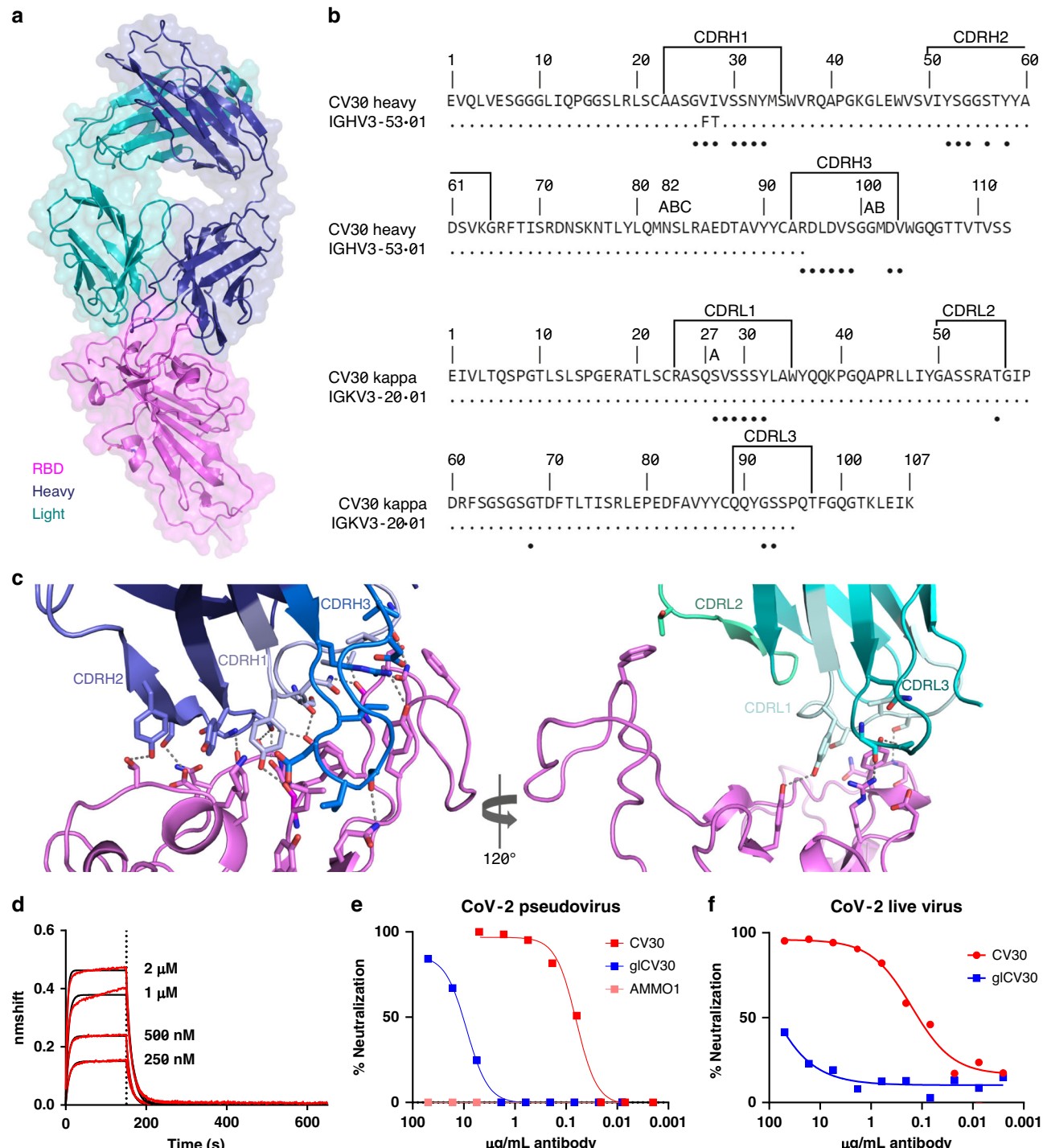

**Fig. 1 Overall structure of CV30 Fab in complex with SARS-CoV-2 RBD and kinetics of glCV30. a** Structure is shown in cartoon with surface representation shown in transparency. CV30 heavy chain is shown in dark blue and light chain in light blue. RBD is shown in pink. **b** Sequence alignment of CV30 heavy and light chains with germline genes. Black circles underneath the sequence indicate residues that interact with the RBD. **c** Details of the interactions of the heavy (left) and light (right) chains with the RBD. Complementary determining regions (CDRs) are labeled and colored as shown. Residues that interacts are shown as sticks and hydrogen bonds are shown in dotted lines. **d** Kinetics of glCV30 binding to RBD measured by BLI. Experimental data are shown in red with fitted curve shown in black. Dashed line represents the transition from association to dissociation phase. **e** CV30 and glCV30 neutralization of SARS-CoV-2 pseudovirus. AMMO1 is an EBV-specific antibody included as a control. **f** CV30 and glCV30 neutralization of SARS-CoV-2 live virus. **e, f** Data points represent the mean of duplicates. Each experiment was repeated two times independently with similar results. **d, e, f**. Source data are provided as a Source Data file.

Mapping the binding interactions of the RBD to each of the antibodies reveals a close overlap in the binding mechanism (Fig. 2d, e). The footprint of the heavy chain is nearly identical, as expected from the shared germline V-gene and sequence similarity. CV30 and CB6 both have longer CDRH3 and bind with higher buried surface area, ~263 and ~251 Å$^2$, respectively, than B38 (~203 Å$^2$) (Fig. 2d and Supplementary Fig. 2). The large difference is in the light chain. CV30 has the smallest binding

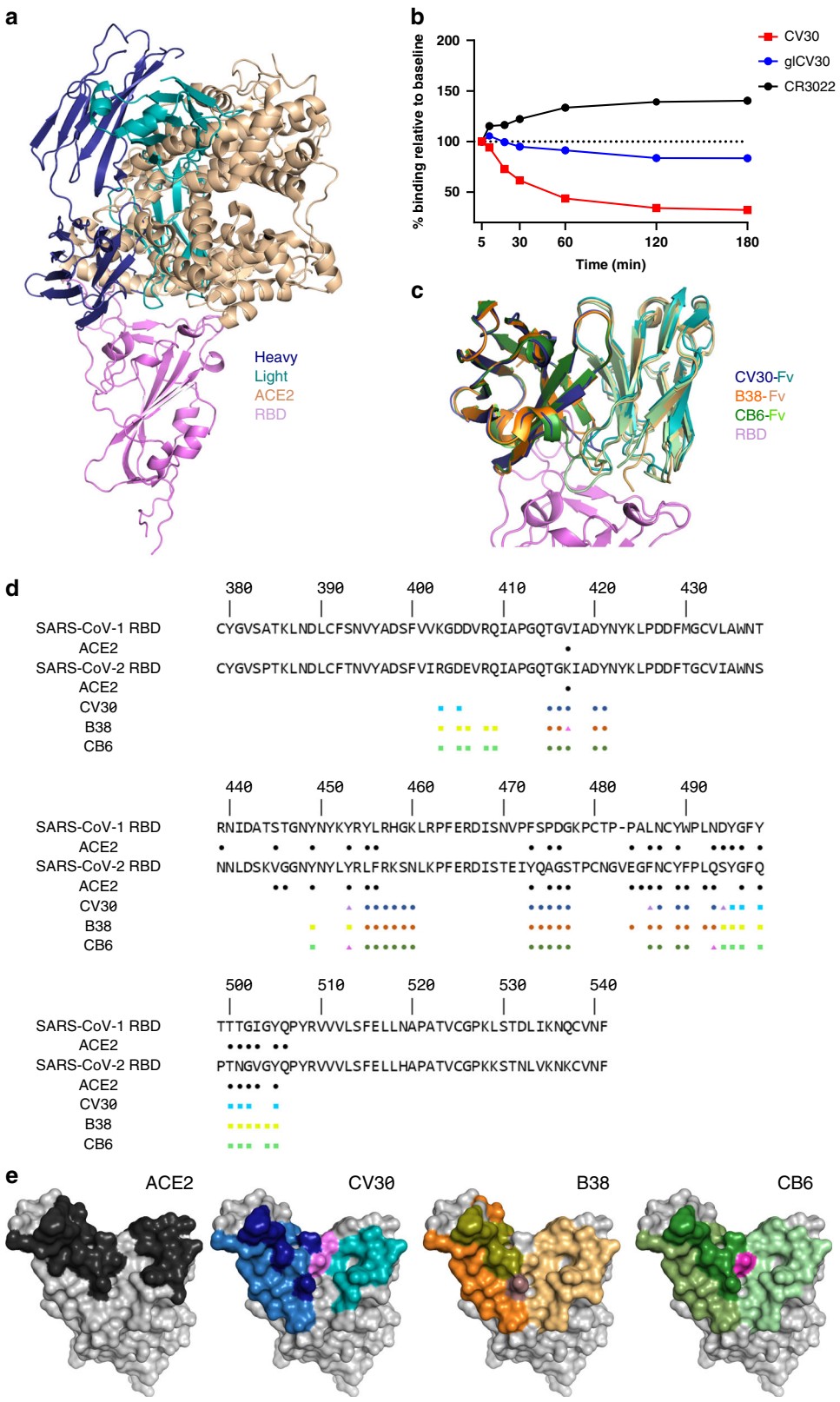

interaction at ~254 Å², B38 has the largest interaction at ~497 Å² and then CB6 at ~354 Å². One of the more interesting findings was the interaction of Thr56 in the CV30 CDRL2, which reaches across the RBD and interacts with Phe486, an interaction that is not found in the other two antibodies (Supplementary Fig. 2). We note that although this manuscript was under revision, two papers were published describing three additional antibodies of

the same heavy chain lineage and binding mechanism[19,20]. Here we demonstrate that this class of SARS-CoV-2 antibody can induce premature S1 shedding.

## Discussion

Our structure indicates that potent neutralizing antibodies against SARS-CoV-2 bind the RBM in the RBD, overlapping the

**Fig. 2 Comparison of the CV30 epitope against ACE2 and other neutralizing antibodies. a** Structural overlay of ACE2/RBD complex with CV30/RBD complex. ACE2 is shown in sand color and RBD is in pink. The heavy chain of CV30 is shown in dark blue and the light chain is in light blue. **b** mAb binding to cell surface expressed SARS-CoV-2 S shows that CV30 induces shedding of the S protein. CR3022 is an RBD-binding antibody that does not induce shedding. Data points represent the mean of duplicates. Each experiment was repeated two times independently with similar results. Source data are provided as a Source Data file. **c** Structural alignment of the variable domains of CV30 (heavy chain is dark blue and light chain is light blue), B38 (heavy chain is dark orange and light chain is light orange), and CB6 (heavy chain is dark green and light chain is light green). **d** Sequence alignment of SARS-CoV-1 RBD and SARS-CoV-2 RBD. The residues that interact with ACE2 are indicated by the black circles. Residues that interact with CV30, B38, and CB6 are indicated by the colored squares (light chain interactions), circles (heavy chain interactions), or triangles (interactions with both chains). **e** Surface representation of the RBD with the binding epitope colored. Light chain interactions are the lightest color, heavy chain interactions are next lightest, and CDRH3 specific interactions are darkest, and interacting with both heavy and light chain is purple.

ACE2-binding site, but recognize residues that are specific for SARS-CoV-2 only, thus explaining the lack of cross neutralization with SARS-CoV-1. Binding of CV30 to cells expressing the SARS-CoV-2 S protein decreased overtime indicating shedding of the S1 subunit as a mechanism of neutralization that complements the steric prevention of S binding to ACE2. It is noteworthy that potently neutralizing antibodies isolated from multiple individuals use the same or similar VH gene to target their epitope, as it has been shown more recently by published studies[12,14,19,20]. In addition, the minimal affinity maturation observed 21 days after infection in the VH gene of CV30 showed ~100–500-fold increase in affinity and neutralization potency, indicating that further affinity maturation may increase potency and potential cross-reactivity. Our studies indicate that the RBD is a promising target for vaccine design, and that these potently neutralizing antibodies should be explored as a treatment for COVID-19 infection.

## Methods

**Recombinant protein expression and purification.** The plasmid encoding the RBD of SARS-CoV-2 spike protein (GenBank: MN908947), residues 319–591, were cloned upstream of a C-terminal HRV3C cleavage site, a monomeric Fc-tag, and an 8×His-tag[5], and was a gift from Dr. Jason McLellan.

One liter of 293SGlycoDelete cells[21] were cultured to a density of 1 million cells/mL and transiently transfected with 500 µg of pαH-RBD-Fc using 2 mg of polyethylenimine (PEI, Polysciences, catalog number 24765). Cultural supernatant was collected 6 days post transfection by centrifugation and sterile filtered using a 0.22 µm vacuum filter. The RBD was purified using protein A agarose resin (GoldBio, catalog number P-400) and cleaving the Fc domain using HRV3C protease (made in house) on-column. The eluate containing the RBD was further purified by size-exclusion chromatography (SEC) using a HiLoad 16/600 Superdex 200 pg column (GE Healthcare) column pre-equilibrated in 2 mM Tris-HCl pH 8.0, 200 mM NaCl. Protein was aliquoted, flash frozen, and stored at −80 °C until needed.

Five hundred milliliters of 293-6E cells (National Research Council of Canada (under license)) were cultured to a density of 1 million cells/mL and transiently transfected with 125 µg each of CV30 Heavy and Kappa chains using 1 mg of PEI. Cultural supernatant was collected 6 days post transfection by centrifugation and sterile filtered using a 0.22 µm vacuum filter. IgG was purified using protein A agarose resin and eluted using Pierce IgG Elution Buffer (Thermo Scientific, catalog number 21004). Eluate was pH adjusted to 7.5 using 1 M HEPES pH 7.5. IgG was further purified by SEC using a HiLoad 16/600 Superdex 200 pg column. Antigen binding fragment (Fab) was generated by incubating IgG with LysC (New England Biolabs, catalog number P8109S) at a ratio of 1 µg LysC per 10 mg IgG at 37 °C for 18 h. Fab unexpectedly stuck to protein A resin and was eluted as mixture of Fab, undigested IgG, and digested Fc product using the IgG elution buffer. Fab and Fc product was purified by SEC. The CV30 Fab and SARS-CoV-2 RBD complex was obtained my mixing Fab and Fc product with a twofold molar excess of RBD and incubated for 90 min at room temperature with nutation followed by SEC. The complex was verified by SDS-polyacrylamide gel electrophoresis analysis.

**Crystal screening and structure determination.** The complex was concentrated to 10 mg/mL for initial crystal screening by sitting-drop vapor-diffusion in the MCSG Crystallization Suite (Anatrace, MCSG-1, MCSG-2, and MCSG-3) using a NT8 drop setter (Formulatrix). Diffracting crystals were obtained in a mother liquor (ML) containing 0.2 M (NH₄) Citrate, tribasic pH 7.0 and 12% (w/v) PEG 3350. The crystals were cryoprotected by soaking in ML supplemented with 30% (v/v) ethylene glycol. Diffraction data were collected at Advanced Photon Source SBC 19-ID at a 12.662 keV. The data set was processed using XDS[22] and data reduction was performed using AIMLESS in CCP4 to a resolution of 2.75 Å. The structure of the complex was solved by molecular replacement using Phaser[23] in Phenix[24] with a search model of SARS-CoV-2 RBD (PDBid: 6LZG)[9] and the Fab

structure (PDBid: 5I1E)[25] divided into Fv and Fc portions. Remaining model building was completed using COOT[26] and refinement was performed in Phenix[24]. The data collection and refinement statistics are summarized in Supplementary Table 1. Structural figures were made in PyMol (Schrodinger, LLC).

**Enzyme-linked immunosorbent assay.** Immulon 2HB microtiter plates (Thermo Scientific) were coated with 50 ng/well of SARS-CoV-1 or SARS-CoV-2 RBD at 4 °C overnight. Plates were washed 4X with phosphate-buffered saline (PBS) with 0.02% Tween-20 (wash buffer). Plates were blocked with 250 µL of 10% non-fat milk and 0.02% Tween-20 in PBS (blocking buffer) for 1 h at 37 °C. After washing 4× with wash buffer, purified IgG was added at an initial concentration of 20 µg/mL and diluted in tenfold serial dilutions in blocking buffer and incubated for 1 h at 37 °C. Plates were washed 4× in wash buffer and the secondary antibody Goat anti-Human Ig-HRP (Southern Biotech, catalog number 2010-05) was added at a 1:3000 dilution and incubated at 37 °C for 1 h. After a final 4× wash, 50 µL of SureBlue Reserve TMB Peroxidase Substrate (SeraCare, catalog number 5120-0081) was added and incubated for 4 min followed by addition of 100 µL of 1 N H₂SO₄ to stop the reaction. The optical density at 450 nm was measured using a SpectraMax M2 plate reader (Molecular Devices). All wash steps were performed using a BioTek 405 Select Microplate Washer.

**Biolayer interferometry.** For kinetic analyses glCV30 was captured on anti-Human IgG Fc capture (AHC) sensors (ForteBio, catalog number 18-5064) at a concentration of 20 µg/mL and loaded for 100 s. After loading, the baseline signal was then recorded for 1 min in KB. The sensors were immersed into wells containing serial dilutions of purified SARS-CoV-2 RBD in KB for 150 s (association phase), followed by immersion in KB for an additional 600 s (dissociation phase). The background signal from each analyte-containing well was measured using VRC01 IgG control reference sensors and subtracted from the signal obtained with each corresponding glCV30 loaded sensor. Kinetic analyses were performed at least twice with an independently prepared analyte dilution series. Curve fitting was performed using a 1:1 binding model and the ForteBio data analysis software. Mean $k_{on}$, $k_{off}$ values were determined by averaging all binding curves that matched the theoretical fit with an $R^2$ value of ≥0.98.

**Pseudovirus neutralization assay.** HIV-1-derived viral particles were pseudo-typed with full-length wild-type SARS-CoV-2 S[27]. Briefly, plasmids expressing the HIV-1 Gag and pol (pHDM540 Hgpm2), HIV-1Rev (pRC-CMV-rev1b), HIV-1 Tat (pHDM-tat1b), the SARS-CoV-2 spike (pHDM-SARS-CoV-2 Spike) and a luciferase/GFP reporter (pHAGE-CMV-Luc2-IRES542 ZsGreen-W) were co-transfected into HEK293T (ATCC, catalog number CRL-3216) cells at a 1:1:1:1.6:4.6 ratio using 293 Free transfection reagent (EMD Millipore, catalog number 72181) according to the manufacturer's instructions. Pseudovirus production was carried out at 32 °C for 72 h after which the culture supernatant was harvested, clarified by centrifugation and frozen at −80 °C.

HEK293T cells stably expressing hACE2 (BEI Resources, catalog number NR-5251) were seeded at a density of $4 \times 10^3$ cells/well in a 100 µL volume in 96-well flat-bottom tissue culture plates. The next day, CV30 and glCV30 were serially diluted (three fold) starting at 50 µg/mL in 30 µL of cDMEM in 96-well round-bottom 27 plates in duplicate or quadruplicate. An equal volume of viral supernatant diluted to result in $2 \times 10^5$ luciferase units was added to each well and incubated for 60 min at 37 °C. Meanwhile 50 µL of cDMEM containing 6 µg/mL polybrene was added to each well of 293T-ACE2 cells (2 µg/mL final concentration) and incubated for 30 min. The media was aspirated from 293T-ACE2 cells and 100 µL of the virus-antibody mixture was added. The plates were incubated at 37 °C for 72 h. The supernatant was aspirated and replaced with 100 µL of Steadyglo luciferase reagent (Promega). 75 µL was then transferred to an opaque, white bottom plate and read on a Fluorskan Ascent Fluorimeter. Control wells containing virus but no antibody (cells + virus) and no virus or antibody (cells only) were included on each plate.

Percent neutralization for each well was calculated as the relative light units (RLU) of the average of the cells + virus wells, minus test wells (cells + mAb + virus), and dividing this result difference by the average RLU between virus control (cells + virus) and average RLU between wells containing cells alone, multiplied by

100. The antibody concentration that neutralized 50% of infectivity (IC50) was interpolated from the neutralization curves determined using the log(inhibitor) vs. response—variable slope (four parameters) fit using automatic outlier detection in Graphpad Prism 8.4.

**Focus reduction neutralization titer assay.** Monoclonal antibodies were serially diluted (three fold) starting at 50ug/mL in serum-free Dulbecco's modified Eagle's medium (DMEM) in duplicate wells and incubated with 100–200 FFU infectious clone derived SARS-CoV-2 virus[28] (kindly provided by V. Menachery and P.Y. Shi) at 37 °C for 1 h. The antibody-virus mixture was added to VeroE6 cell (C1008, ATCC, #CRL-1586) monolayers and incubated at 37 °C for 1 h. Post incubation, the antibody-virus mixture was removed and 100 μL of prewarmed 0.85% methylcellulose (Sigma, #M0512-250) was added to each well. Plates were incubated at 37 °C for 24 h. After 24 h, methylcellulose overlay was removed, cells were washed twice with PBS and fixed with 2% paraformaldehyde in PBS for 30 min at room temperature. Following fixation, plates were washed twice with 1x PBS and 100 μl of permeabilization buffer (0.1% BSA-Saponin in PBS) (Sigma Aldrich), was added to the fixated Vero cell monolayer for 20 minutes. Cells were incubated with an anti-SARS-CoV spike protein primary antibody conjugated to biotin (CR3022-biotin) for 1 hour at room temperature, then with avidin-HRP conjugated secondary antibody for 1 hour at room temperature. Foci were visualized using True Blue HRP substrate and imaged on an ELISPOT reader (CTL). Foci were visualized using an ELISPOT reader (CTL ImmunoSpot S6 Universal Analyzer) and enumerated using Viridot[29]. The average number of foci in the virus only sample (duplicate) were used to calculate the neutralization curves: 1 − (ratio of mean number of foci in the presence of mAb and virus only control). The antibody concentration that neutralized 50% of infectivity (IC50) was interpolated using a four-parameter nonlinear regression in GraphPad Prism 8.4.

**mAb binding to cell surface expressed SARS-CoV-2 S over time**. cDNA for the full-length SARS-CoV-2 S isolate USA-WA1/2020 was codon optimized and synthesized by Twist Biosciences and cloned into the pTT3 vector using InFusion cloning (Clontech). pTT3-SARS-CoV-2-S was transfected into HEK-293-6E cells (National Research Council of Canada, under license) using 293 Free transfection reagent according to the manufacturer's instructions. Transfected cells were incubated for 24 h at 37 °C with shaking. The next day, 200,000 cells in 60 μL of FreeStyle medium containing 10% fetal bovine serum (FBS) and 1% Pen/Strep were added to a round-bottom 96-well plate. 300 nM mAb in FreeStyle medium and the cells were equilibrated separately for 15 min to 37 °C. At each timepoint (180, 120, 60, 30, 20, 10, 5 min), 30 μL of each mAb were mixed with the cell suspension to a final concentration of 100 nM mAb. After incubation for the allotted time, the samples were placed on ice and washed twice with ice-cold FACS buffer (PBS + 2% FBS + 1 mM EDTA) before staining cells with PE-conjugated AffiniPure Fab fragment goat anti-human IgG (1 : 50 dilution, Jackson Immunoresearch, catalog number 109-117-008) for 30 min on ice in the dark. Cells were washed once with FACS buffer and stained with Fixable Viability Dye eFluor 506 (1 : 200 dilution, eBioscience, catalog number 65-0866-14) for 20 min on ice in the dark. Cells were washed once with FACS buffer, fixed with 10% formalin for 15 min on ice in the dark, and resuspended in 200 μL of FACS buffer to be analyzed by flow cytometry using a LSRII (BD). Mean fluorescence intensity (MFI) for each sample was determined at each timepoint and each sample was normalized to the MFI at the 5 min timepoint (MFI/MFI 5 min × 100).

**Reporting summary**. Further information on research design is available in the Nature Research Reporting Summary linked to this article.

## Data availabilty

Coordinates and structure factors for CV30 Fab-SARS-CoV-2 RBD complex have been deposited in the Protein Data Bank (PDB) under the accession code 6XE1. In addition, the following publicly available datasets are mentioned: PDB: 6LZG[9], PDB: 7BZ5[14], CB6 (PDB: 7C01)[12], and PDBid: 5I1E[25]. All reagents generated in this study are available upon request through Material Transfer Agreements. pTT3-derived plasmids and 293-6E cells require a license from the National Research Council (Canada). Source data are provided with this paper.

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

## Acknowledgements

This work was supported by generous donations to Fred Hutch COVID-19 Research Fund. We thank Dr. McLellan for providing the SARS-CoV-2 RBD plasmid. We thank the J. B. Pendleton Charitable Trust for its generous support of Formulatrix robotic instruments. Results shown in this report are derived from work performed at Argonne National Laboratory, Structural Biology Center (SBC), ID-19, at the Advanced Photon Source. SBC-CAT is operated by UChicago Argonne, LLC, for the U.S. Department of Energy, Office of Biological and Environmental Research under contract DE-AC02-06CH11357. The SARS-CoV-2 neutralization assay efforts were in part supported by the Emory EVPHA Synergy Fund award, Center for Childhood Infections and Vaccines, Children's Healthcare of Atlanta, COVID-Catalyst-I3 Funds from the Woodruff Health Sciences Center and Emory School of Medicine.

## Author contributions

N.K.H., A.T.M., L.S., and M.P. conceived the project. N.K.H., E.S., M.S., A.T.M., L.S., and M.P designed the experiments. J.F. cloned the plasmids. N.K.H. and A.B.S. expressed and

purified the proteins. N.K.H. crystallized proteins, collected and processed the diffraction data, and solved the crystal structure. N.K.H. performed kinetic experiments. V.V.E. and Y.-H.W. performed neutralization assays. E.S. performed shedding experiments. N.K.H., A.T.M., L.S., and M.P. analyzed and discussed data. N.K.H. and M.P. wrote the original manuscript draft. N.K.H., A.T.M., L.S., and M.P. reviewed and edited the manuscript.

## Competing interests

The authors declare no competing interests. A provisional patent application (U.S. Provisional Application number 63/016268) has been filed on the SARS-CoV-2-specific monoclonal antibodies isolated herein.
