## [Peer Review File · Nature Communications]

REVIEWER COMMENTS

Reviewer #1 (Remarks to the Author):

In this manuscript, Hurlburt et al. determined the crystal structure of the complex of SARS-CoV-RBD bound by a neutralizing antibody CV30. It was found that the binding mode of CV30 was similar to those of two other published antibodies (CB6 and B38) through sequence and structural comparisons. The authors also showed that the CV30 epitope largely overlaps with the ACE2-binding site on the RBD, thus providing the structural basis for its neutralization by ACE2 engagement competition. My major concern of this short report is the lack of further biochemistry and virology studies after structural determination and incompleteness in elucidating the neutralization mechanism. The authors used mutagenesis, binding assay and neutralization of pseudovirus to reveal the importance of V27 and I28 in the HCDR1 for binding and neutralization. However, considering the very low hypermutation rate, it is expected that the authors performed similar studies on other HCDR1 and HCDR2 residues to reveal a comprehensive structure-function relationship of the interface. A head-to-head comparison of the detailed interactions around the HCDR3 is also required due to sequence variations here. The neutralization of live virus data should also be included in the study procedure, at least for important residues on the antibody for binding and neutralization of pseudovirus.

It is evident from the structural comparison that CV30 would compete ACE2 binding, one important aspect of its neutralization. However, I think that the effect of neutralizing antibodies after binding to the "up" or "down" RBD would be more complicated than just receptor competition. Previous studies have shown that antibody binding would induce spike prefusion to postfusion transition and inactivate the spike on the viron. It was also recently reported that the shedding of the S1 subunit by some SARS-CoV-2 antibody may play a significant role in neutralization. A more comprehensive study of CV30 neutralization is expected.

Several minor issues:

1. Fig. 1b was not quoted in the manuscript.
2. Abstract is too simple.
3. Line 37: 0.03ug/mL represents IC50?
4. Line 90: CDRK2 should be consistent with CDRL2 in Fig. 1b.

Reviewer #2 (Remarks to the Author):

Hurlburt et al. reported a potent neutralizing monoclonal antibody against SARS-CoV-2, CV30, that was isolated from a patient infected with SARS-CoV-2. The crystal structure of CV30/RBD complex has also been solved and the binding of germline format was evaluated. In general, the results are interesting. However, I have some concerns so cannot recommend for acceptance for publication right now.

1. The authors only used pseudovirus to test the neutralizing activity. The test in authentic virus is necessary.
2. In Figure 1e, the error bars of neutralization curve of CV30 look very high, which makes the results unreliable.
3. In Figure 2c, the authors used SARS-CoV-1 RBD, while SARS-CoV RBD was used in figure legend. SARS-CoV-1 and SARS-CoV should be uniformed.
4. The authors showed the difference of binding sites between SARS-CoV and SARS-CoV-2 to explain the lack of cross neutralization of CV30 with SARS-CoV. It is better to add an experiment (e.g, ELISA)

to show the binding activities of CV30 to SARS-CoV RBD and SARS-CoV-2 RBD respectively.

5. The authors thought "It is noteworthy that potently neutralizing antibodies isolated from multiple individuals use the same or similar VH gene to target their epitope." Here only three antibodies were used for comparison. The evidence is not strong enough for supporting the conclusion.

Response to reviewer's comments

Reviewer #1 (Remarks to the Author):

In this manuscript, Hurlburt et al. determined the crystal structure of the complex of SARS-CoV-RBD bound by a neutralizing antibody CV30. It was found that the binding mode of CV30 was similar to those of two other published antibodies (CB6 and B38) through sequence and structural comparisons. The authors also showed that the CV30 epitope largely overlaps with the ACE2-binding site on the RBD, thus providing the structural basis for its neutralization by ACE2 engagement competition. My major concern of this short report is the lack of further biochemistry and virology studies after structural determination and incompleteness in elucidating the neutralization mechanism.

We thank the reviewer for reviewing our short report and for raising major concerns that we hope to have addressed in this revision.

The authors used mutagenesis, binding assay and neutralization of pseudovirus to reveal the importance of V27 and I28 in the HCDR1 for binding and neutralization. However, considering the very low hypermutation rate, it is expected that the authors performed similar studies on other HCDR1 and HCDR2 residues to reveal a comprehensive structure-function relationship of the interface. A head-to-head comparison of the detailed interactions around the HCDR3 is also required due to sequence variations here.

We have indeed used mutagenesis to study the function of the two residues in the CDRH1 that were mutated from germline in the mature CV30. We have not performed similar studies on other residues since they are not mutated from germline. We feel that further mutational analysis, while interesting from a structure-function standpoint, would not answer any questions directly involved in the role of affinity maturation which we meant to address here.

The neutralization of live virus data should also be included in the study procedure, at least for important residues on the antibody for binding and neutralization of pseudovirus.

We thank the reviewer for the suggestion to add CV30 and gICV30 neutralization of live virus, which we have performed and added to the revised manuscript. We have now added the data on page 4 as follows: "We also performed live virus neutralization assay. Compared to the pseudovirus assay, CV30 neutralized the live virus less potently, with an IC50 of 0.118 µg/mL while gICV30 failed to achieve 50% neutralization (Fig. 1f)."

It is evident from the structural comparison that CV30 would compete ACE2 binding, one important aspect of its neutralization. However, I think that the effect of neutralizing antibodies after binding to the "up" or "down" RBD would be more complicated than just receptor competition. Previous studies have shown that antibody binding would induce spike prefusion to postfusion transition and inactivate the spike on the viron. It was also recently reported that the shedding of the S1 subunit by some SARS-CoV-2 antibody may play a significant role in neutralization. A more comprehensive study of CV30 neutralization is expected.

To address the neutralization mechanism, we performed the S1 shedding assay described in (DOI: 10.1126/science.abc7424). CV30 showed decreased binding to 293 cells expressing full-length SARS-CoV-2 S protein over time indicating the shedding of the S1 subunit. This decrease in binding was not observed with the anti-RBD CR3022 antibody control, which was shown to not induce shedding in the above reference. We have now added this in Fig 2b and the description on page 5 of the revised manuscript.

Several minor issues:

1. Fig. 1b was not quoted in the manuscript.
2. Abstract is too simple.
3. Line 37: 0.03ug/mL represents IC50?
4. Line 90: CDRK2 should be consistent with CDRL2 in Fig. 1b.

We thank the reviewer for these observations. They have been corrected/addressed in the revised manuscript (see track changes).

Reviewer #2 (Remarks to the Author):

Hurlburt et al. reported a potent neutralizing monoclonal antibody against SARS-CoV-2, CV30, that was isolated from a patient infected with SARS-CoV-2. The crystal structure of CV30/RBD complex has also been solved and the binding of germline format was evaluated. In general, the results are interesting. However, I have some concerns so cannot recommend for acceptance for publication right now.

1. The authors only used pseudovirus to test the neutralizing activity. The test in authentic virus is necessary.

We have now performed the live virus neutralization for CV30 and glCV30 and have now added the data as follows on page 4: “We also performed live virus neutralization assay. Compared to the pseudovirus assay, CV30 neutralized the live virus less potently, with an IC₅₀ of 0.118 µg/mL while glCV30 failed to achieve 50% neutralization (Fig. 1f).”

2. In Figure 1e, the error bars of neutralization curve of CV30 look very high, which makes the results unreliable.

We have repeated the pseudovirus neutralization assay with virus produced at 32°C. The IC₅₀ of CV30 agrees well with our previously published report using pseudovirus produced at 37°C but with much smaller error bars, and increased precision (see revised fig. 1e).

3. In Figure 2c, the authors used SARS-CoV-1 RBD, while SARS-CoV RBD was used in figure legend. SARS-CoV-1 and SARS-CoV should be uniformed.

We have adopted the SARS-CoV-1 naming throughout the manuscript.

4. The authors showed the difference of binding sites between SARS-CoV and SARS-CoV-2 to explain the lack of cross neutralization of CV30 with SARS-CoV. It is better to add an experiment (e.g, ELISA) to show the binding activities of CV30 to SARS-CoV RBD and SARS-CoV-2 RBD respectively.

We thank the reviewer for this comment. The binding of CV30 to SARS-CoV-1 vs SARS-CoV-2 S2P was done in our previous manuscript (DOI:<https://doi.org/10.1016/j.immuni.2020.06.001>) and CV30 did not bind SARS-CoV-1 S2P, indicating a lack of cross-reactivity. We replicated these findings using ELISA against the RBD from SARS-CoV-1 and SARS-CoV-2 (extended data fig. 1) and included them on page 4.

5. The authors thought “It is noteworthy that potently neutralizing antibodies isolated from multiple individuals use the same or similar VH gene to target their epitope.” Here only three antibodies were used for comparison. The evidence is not strong enough for supporting the conclusion.

Since the submission of our manuscript, additional manuscripts have been published reporting the same observation. We have now added these reports, published in Science and Cell, to our list of references. Thus, we believe that this observation warrants being mentioned in this manuscript.

REVIEWERS' COMMENTS

Reviewer #1 (Remarks to the Author):

The authors have thoroughly addressed my concerns and included additional experimental data as suggested. I am happy to recommend publication.